# Development and Characterization of Oral Raft Forming In Situ Gelling System of Neratinib Anticancer Drug Using 3^2^ Factorial Design

**DOI:** 10.3390/polym14132520

**Published:** 2022-06-21

**Authors:** Umme Hani, Mohamed Rahamathulla, Riyaz Ali M. Osmani, M.Yasmin Begum, Shadma Wahab, Mohammed Ghazwani, Adel Al Fatease, Ali H. Alamri, Devegowda V. Gowda, Ali Alqahtani

**Affiliations:** 1Department of Pharmaceutics, College of Pharmacy, King Khalid University (KKU), Abha 62529, Saudi Arabia; rahapharm@gmail.com (M.R.); yaminimp47@gmail.com (M.Y.B.); myghazwani@kku.edu.sa (M.G.); afatease@kku.edu.sa (A.A.F.); aamri@kku.edu.sa (A.H.A.); 2Department of Pharmaceutics, JSS College of Pharmacy, JSS Academy of Higher Education and Research (JSS AHER), S.S. Nagara, Mysuru 570015, Karnataka, India; riyazosmani@gmail.com (R.A.M.O.); dvgowda@jssuni.edu.in (D.V.G.); 3Department of Pharmacognosy, College of Pharmacy, King Khalid University, Abha 62529, Saudi Arabia; shad.nnp@gmail.com; 4Cancer Research Unit, King Khalid University, Abha 62529, Saudi Arabia; 5Department of Pharmacology, College of Pharmacy, King Khalid University, Abha 62529, Saudi Arabia; amsfr@kku.edu.sa

**Keywords:** neratinib, breast cancer, 3^2^ factorial design, HPMC K4M, sodium alginate, oral raft, in situ gelling system

## Abstract

Neratinib (NTB) is an irreversible inhibitor of pan-human epidermal growth factor receptor (HER-2) tyrosine kinase and is used in the treatment of breast cancer. It is a poorly aqueous soluble drug and exhibits extremely low oral bioavailability at higher pH, leading to a diminishing of the therapeutic effects in the GIT. The main objective of the research was to formulate an oral raft-forming in situ gelling system of NTB to improve gastric retention and drug release in a controlled manner and remain floating in the stomach for a prolonged time. In this study, NTB solubility was enhanced by polyethylene glycol (PEG)-based solid dispersions (SDs), and an in situ gelling system was developed and optimized by a two-factor at three-level (3^2^) factorial design. It was analyzed to study the impact of two independent variables viz sodium alginate [A] and HPMC K4M [B] on the responses, such as floating lag time, percentage (%) water uptake at 2 h, and % drug release at 6 h and 12 h. Among various SDs prepared using PEG 6000, formulation 1:3 showed the highest drug solubility. FT-IR spectra revealed no interactions between the drug and the polymer. The percentage of drug content in NTB SDs ranged from 96.22 ± 1.67% to 97.70 ± 1.89%. The developed in situ gel formulations exhibited a pH value of approximately 7. An in vitro gelation study of the in situ gel formulation showed immediate gelation and was retained for a longer period. From the obtained results of 3^2^ factorial designs, it was observed that all the selected factors had a significant effect on the chosen response, supporting the precision of design employed for optimization. Thus, the developed oral raft-forming in situ gelling system of NTB can be a promising and alternate approach to enhance retention in the stomach and to attain sustained release of drug by floating, thereby augmenting the therapeutic efficacy of NTB.

## 1. Introduction

Colon cancer, breast cancer, and lung cancer are the most common cancers affecting women, accounting for almost 50%. Breast cancer is the most commonly diagnosed cancer worldwide, accounting for around 30% of all common cancers [1,2,3,4]. Breast cancer is usually diagnosed in one among eight women worldwide. Approximately 90–95% of breast cancer cases are due to sedentary lifestyle and environmental factors, and around 5–10% of breast cancer cases are due to genetic disorders [5,6]. The core origin of this cancer is the lining of epithelial cells in the terminal duct of the lobule [7]. 

Neratinib (NTB), 6,7-disubstituted-4-anilinoquinoline-3-carbonitrile, is an irreversible inhibitor of pan-human epidermal growth factor (HER-2) receptor tyrosine kinase, thereby decreasing autophosphorylation. When NTB is used as a treatment for breast cancer, it reduces proliferation of cells by inhibiting the cell cycle regulatory pathway and downstream signal transduction events, causing arrest at Gap 1/DNA synthesis. However, the aqueous solubility of NTB is poor, and it exhibits the least oral bioavailability at alkaline pH, leading to a diminished therapeutic efficacy in GIT [8,9]. Drugs exhibiting stability issues at higher pH can be formulated as gastro-retentive drug delivery systems. These systems are designed to release the drug in a sustained manner by letting it remain floating in the stomach. A novel drug delivery, raft-forming systems have gained considerable interest worldwide. Raft formation results from the formation of cohesive viscous gel when formulation meets gastric fluid, resulting in the swelling of each portion of liquid, forming a continuous layer known as a raft. Due to its low bulk density because of the release of CO_2_, the raft floats on gastric fluids. The formulation comprises a gelling agent such as carbopol, HPMC, xanthan gum, sodium alginate, etc., and alkaline bicarbonate, which results in the formation of CO_2_ to make the formulation float on the gastric fluid [10,11,12]. 

Interestingly, there are no reports of the oral raft-forming in situ gelling system of NTB with sodium alginate or HPMC-K4M alone as a polymer, making the current work unique. The efficacy of the raft-forming in situ gelling system of some poorly soluble anticancer drugs, such as doxorubicin, paclitaxel, and curcumin, has been shown to enhance the solubility and bioavailability of in vitro dissolution and anticancer therapy [13,14,15]. As a result, a raft-forming in situ gelling floating system was developed to improve gastric retention and drug release in a controlled manner by allowing it to float in the stomach for several hours. The developed raft-forming in situ gel was optimized by using a two-factor at three-level (3^2^) factorial design and was analyzed to study the impact of two independent variables [16], sodium alginate [A] and HPMC K4M [B], on the responses, such as floating lag time, % water uptake at 2 h, and % drug release at 6 h and 12 h. First we evaluated the solid dispersions of NTB and saturation solubility, conducted FT-IR studies and scanning electron microscopy, and then carried out physicochemical evaluations of the raft-forming gel, an in vitro gelling study, an in vitro floating study, a water uptake study, and in vitro drug release and release kinetics studies.

## 2. Resources and Methods 

### 2.1. Materials

Neratinib was obtained from Mesochem Technology Co., Ltd., Beijing, China. Sodium alginate, sodium citrate, sodium bicarbonate, and polyethylene glycol (PEG) were procured from Sigma Aldrich, Mumbai, India. HPMC K4M was purchased from Parchem, New Rochelle, NY, USA. All other reagents and chemicals employed for the study were of pharmaceutical grade. 

### 2.2. Methods

#### 2.2.1. PEG—Neratinib Solid Dispersion Preparations

The solvent evaporation method was adopted to prepare solid dispersions of NTB and PEG-6000 as a carrier with a 1:1, 1:2, and 1:3 ratio. A total of 20 mL methanol was taken in a 100 mL beaker, 100 mg of the drug was made to dissolve, and then PEG was added and dissolved to obtain a clear solution. The solvent was made to evaporate by heating at 60 °C in a water bath (GFL Gasellschaft fur Labortechnik-Model 1042, Burgwedel, Germany). The resulting solid dispersion was stored in a desiccator to get constant weight, crushed, and sifted through a No.20 sieve (841 microns) and utilized for further evaluations [17].

#### 2.2.2. Determination of Saturation Solubility

The enhancement of the solubility of NTB by forming solid dispersions was determined by measuring the saturation solubility. The excess known quantity of the solid dispersion of NTB was added to 100 mL 0.1N HCL (hydrochloric acid). The mixture was stirred continuously for 48 h, maintaining a temperature of 37 ± 0.5 °C with a water bath. Then the sample was filtered and analyzed for the drug concentration after suitable dilutions using a UV-visible spectrophotometer at 265 nm [18,19,20].

#### 2.2.3. Analytical Method Development

##### Determination of Absorption Maxima of NTB

The lambda max of the drug was determined using a UV-visible spectrophotometer (UV-1800, Shimadzu, Kyoto, Japan). Stock solution (1000 µg/mL) was prepared by weighing 100 mg of NTB in a 100 mL volumetric flask. The drug was made to dissolve by using 1 mL methanol to make up the volume with 0.1 N HCL. Using the stock solution, serial dilutions with a concentration of 2, 4, 6, 8, and 10 µg/mL were prepared and the resulting solutions were scanned between 200 and 400 nm using 0.1 N as a blank, and the lambda max of the NTB was determined. A calibration curve was plotted by taking a known concentration (2–30 µg/mL) and absorbance. With the selected concentration range, desirable linearity was obtained with a correlation coefficient of 0.9991 (R^2^). With the obtained concentration range, the std graph obeyed the Beer–Lamberts law.

#### 2.2.4. Fourier Transform Infrared Spectroscopy (FT-IR) Studies

FT-IR spectra of NTB and solid dispersions were obtained using an FT-IR spectrophotometer (Model 4700, Thermo Fisher Scientific, Waltham, MA, USA). The samples were prepared by the KBr disk method (3 mg of sample in 300 mg KBr) using a hydrostatic press in the scanning range of 400–4000 cm^−1^ and a resolution of 2 cm^−1^ [17].

#### 2.2.5. Scanning Electron Microscopy

The morphology was analyzed by SEM micrographs of NTB and NTB-PEG solid dispersion using a scanning electron microscope (Zeiss, EVO LS, Smart SEM 5.05, Jena, Germany) at an acceleration voltage of 20 kV and a magnification of 500× at room temperature. 

#### 2.2.6. NTB Floating In Situ Gel Preparation

All the additives used in the preparations were passed from a No. 60 sieve (250 microns). Required ingredients for the preparation, like sodium alginate, HPMC K4M, sodium bicarbonate, and sodium citrate, were accurately weighted as per the formulation chart depicted in Table 1. HPMC K4M was dissolved using 40 mL of deionized water. The required quantity of sodium bicarbonate and sodium citrate were incorporated in it while stirring to attain complete homogenous dispersion. An NTB PEG solid dispersion equivalent to 100 mg of drug was dissolved in the solution. Sodium alginate was dissolved using deionized water (30 mL) taken in a beaker pre-heated to around 60 °C on a hot plate (Whirlmatic Spectra Lab, Mumbai, India) with continuous stirring. The sodium alginate solution was cooled to 40 °C and added to the HPMC K4M solution. The total amount of the preparation finally reached 100 mL, making use of distilled water after adding methyl paraben as a preservative and mixed thoroughly [21].

#### 2.2.7. Experimental Design

The interaction and relationship between dependent and independent variables can be studied using a scientific and systemic approach, i.e., experimental design. The optimization of the formulations was performed using 3^2^ factorial design. The effect of the independent variables and their interactions can be determined from the chosen experimental design, which can provide a satisfactory degree of freedom. Two independent factors (variables), sodium alginate (A) and HPMC K4M (B), were selected and evaluated at two levels, i.e., higher level (−1), medium level (0), and lower level (+1). The responses (independent variables) chosen to know the effect of the factors were floating lag time, % water uptake at 2 h, and % drug release at 6 h and 12 h [6]. The analysis of the obtained data was carried out employing Design-Expert software (version 130.2.0) offered from Stat-Ease Inc., Minneapolis, MN, USA. Table 2 enlist the factors and their levels for preparing the oral raft-forming in situ gelling system of NTB. 

### 2.3. Evaluation

#### 2.3.1. Physicochemical Properties

The prepared floating in situ gel formulations were examined for color, taste, and odor using natural senses [22,23,24].

#### 2.3.2. Drug Content Determination

The required quantity of the solid dispersion formulation was measured, which was equivalent to 100 mg of the drug, and was placed in a volumetric flask of 100 mL. The drug content was measured at the absorption maxima of 265 nm using a UV-visible spectrophotometer (UV-1800, Shimadzu, Japan) [25]. 

#### 2.3.3. pH Determination

All the developed formulations were tested for pH using a previously calibrated digital pH meter (Mettler Toledo MP 220, Greifensee, Switzerland) by placing the sensor end in the prepared formulation. The study was conducted at room temperature [26].

#### 2.3.4. In Vitro Gelling Capacity

In vitro gelation of the prepared formulations was measured as per the method reported in our previously published work [7]. One mL of the precisely measured colored formulation was placed in a test containing 5 mL of 0.1 N HCL with a pH 1.2 and was maintained at body temperature (37 ± 0.5 °C) with slight stirring to avoid breaking the formed gel. Based on the gelation time, stiffness of the gel formed, and the gel stability in a test tube, the gelling capacity was categorized as follows: no gelation (−), gelation after a few min following quick dispersion (+), instant gelation retained for few hours (++), and instant gelation retained for a longer period of time (+++).

#### 2.3.5. In Vitro Floating Study

The study was performed by placing 10 mL of the prepared formulation in a beaker containing 0.1N HCL (900 mL) with a pH of 1.2 (determined used digital pH meter), and the temperature was maintained at 37 ± 0.5 °C. The beaker was placed to avoid turbulence or any disturbance. The length of time for the liquid formulation to float on the surface of the specified medium was noted as the floating lag time, and the total time required by the formed gel to float on the medium was termed the floating duration. In addition, the floating lag time and floating duration were determined for all the formulations (F1–F8) [13].

#### 2.3.6. Water Uptake Study

Determination of the water uptake was carried out via a method previously reported by Jahangir et al. [27]. The formulation was placed in 40 mL of 0.1 N hydrochloric acid of pH 1.2 at a temperature of 37 ± 0.5 °C. The formed gel portion was separated from the medium. The excess medium embedded in the gel was removed using tissue paper. The initial weight of the gel was measured as W1, and then the gel was placed in 10 mL of distilled water. After a 30 min time interval the weight of the gel was measured again after decanting excess water and was noted as W2. The difference between W2 and W1 was determined as follows:%Water uptake =W2 − W1/W2 × 100(1)

#### 2.3.7. In Vitro Drug Release Study

The NTB release study was carried out in triplicate using a USP Type II (paddle type) dissolution apparatus (model). The stirring speed of 50 rpm was fixed, which is believed to simulate the mild agitation in vivo and avoid breaking the formation of the in situ gel. Five mL of the prepared formulation was incorporated into 900 mL dissolution medium of pH 1.2 (0.1N HCl) and the temperature was maintained at 37 ± 0.5 °C. About 5 mL of the samples were withdrawn at 1, 2, 4, 6, 8, 10, and 12 h and replaced by an equal volume of fresh dissolution medium immediately, which was maintained at the same temperature to maintain sink condition. The aliquot samples were filtered using a 0.45 μ membrane filter and analyzed using a UV-visible spectrophotometer (UV-1800, Shimadzu, Kyoto, Japan) at 265 nm after suitable dilutions. The dissolution profile of the raft formulation was obtained by plotting a graph taking cumulative drug release on the y-axis and time on the X-axis [28,29,30].

#### 2.3.8. Kinetics of Drug Release Studies

To determine the kinetics of the drug release, the dissolution profile of each batch was adapted for different models, including first-order, zero-order, Hixon and Crowell, Higuchi, and Korsemeyer Peppas (KP). The KP equation describes the method to explain the drug release mechanisms [31,32,33].

## 3. Results and Discussion

The present study was aimed at enhancing the aqueous solubility of NTB by solid dispersion and developing an in situ gelling raft-forming floating system to improve gastric retention and drug release in a controlled manner by remaining floating in the stomach. NTB is an irreversible inhibitor of pan-human epidermal growth factor (HER-2) receptor tyrosine kinase. The lambda max of the anticancer drug NTB was observed to be 265 nm and further used for the study. The aqueous solubility of NTB is poor and it exhibits extremely low oral bioavailability at alkaline pH, leading to diminished therapeutic efficacy in GIT. The solubility of NTB is enhanced by the solid dispersion method and formulates an oral raft-forming in situ gelling system of NTB, an anticancer drug.

### 3.1. Saturation Solubility Studies of NTB PEG Solid Dispersions

The results of the saturation solubility of free NTB and NTB PEG solid dispersions in 0.1 N hydrochloric acid (pH 1.2) are depicted in Table 3. NTB was sparingly soluble in water. NTB solid dispersions enhanced the saturation solubility of the drug, which may have been due to the formation of complexity between the drug and carrier. Because of the solid dispersions, particle size reduction took place, leading to an enhanced surface area, thereby enhancing the dissolution rate. Among all solid dispersion, A3 (NTB:PEG ratio 1:3) exhibited the highest solubility (132.11 ± 2.14 μg/mL), which was used for further study. 

### 3.2. FT-IR Spectra

The FT-IR spectra of NTB and NTB PEG solid dispersion are shown in Figure 1. The spectra of NTB revealed characteristic absorption peaks at 3034, 3002, 2930, and 2850 cm^−1^ (for C–H stretching vibrations); 3430 cm^−1^ (for N–H stretching vibrations); 2206 cm^−1^ (for C≡N stretching vibrations); 1686 cm^−1^ (for C=O stretching vibrations); 1496 cm^−1^ (C=C stretching vibrations); and 1286 cm^−1^ and 1269 cm^−1^ (for C-H bending vibrations), and strong peaks were observed between 1500 and 1600 cm^−1^, representing the existence of an aromatic ring. In the spectra of NTB PEG solid dispersion, all the peaks corresponding to the drug were observed with no significant changes, indicating no specific interaction between the polymer and drug.

### 3.3. Scanning Electron Microscopy

The surface morphology was analyzed by SEM micrographs of NTB and NTB–PEG solid dispersion, as shown in Figure 2. Pure NTB appeared crystalline, whereas PEG–NTB solid dispersion was deemed to be amorphous powder. From the obtained micrographs, it was evident that after solid dispersion of the drug along with PEG there was a reduction in particle size and disappearance of the crystallinity of the drug. Due to the presence of carrier PEG, the surface of the NTB solid dispersion looked smooth. 

### 3.4. Drug Content

The percentage drug content in NTB solid dispersion ranged from 96.22 ± 1.67% to 97.70 ± 1.89%, as mentioned in Table 3, indicating uniform distribution of NTB in the prepared solid dispersions. 

### 3.5. Physicochemical Properties

The prepared formulations (F1–F9) using different sodium alginate and HPMC K4M concentrations were clear, devoid of any turbidity, had suspended particles with a bland taste, and had no characteristic odor. 

### 3.6. pH Determination

The prepared formulations (F–F9) exhibited pH values ranging from 7.1 ± 0.21 to 7.8 ± 0.24, as shown in Table 4, and showed various evaluation parameters of an in situ gelling system. All the formulations were slightly alkaline. In a simulated gastric environment, all the prepared formulations exhibited instantaneous sol-to-gel transition. The resulting pH range of formulations was suitable to retain a liquid state. 

### 3.7. In Vitro Gelling Capacity

All the prepared formulations resulted in immediate gelation that was retained for an extended period. The systems that resulted in instantaneous gel formation upon exposure to biological fluids and body temperature are ideal in situ gelling systems. As the concentration of HPMC K4M increased, gelation was observed to be enhanced. Formulations with the lowest concentration of polymer resulted in weak gel formation, which may not be able to withstand peristaltic waves of the GIT. Hence, an optimum polymer concentration was required to get an ideal gelling system. 

### 3.8. In Vitro Floating Study

When the system is buoyant, drug can be released at a desired rate and, in turn, diminish the side effects of the drug, such as gastric ulceration, by avoiding direct contact with the stomach mucosa [22]. The time required by the system to float on the surface of the medium is termed floating lag time, which is the preliminary measure of the floating performance of the formulation. The duration of floating is the total period of floating of the formulation on the surface of the medium. In vitro floating is an obligatory parameter to be assessed prior to the assessment of the formulation in vivo [24]. From our previously published study, it was observed that the excess concentration of sodium bicarbonate that has been incorporated as a gas-forming agent to achieve buoyancy decreased the floating lag time and floating duration of the formulations. In the present study, the previously optimized concentration of sodium bicarbonate (1%) was used along with sodium citrate (1%), which was used to maintain fluidity of the formulation prior to administration. Sodium alginate and HPMC K4M were used as gelling agents [13]. All the developed formulations were shown to float on the surface of 0.1N HCL (simulated gastric fluid), but formulations with a higher concentration of HPMC K4M (F4–F9) showed a floating lag time of more than 30 min and the formulations F1, F2, and F3 showed a floating lag time of 2.4, 3.1, and 6.2 min, respectively. The results of floating duration were different for all the formulations. Formulation F1 remained floating for more than 24 h, whereas F2, F3, F5, and F6 remained buoyant for less than 12 h. The floating duration of formulations F7 to F9 was less than 3 h, which may have been due to the higher concentration of HPMC K4M. F4 showed a floating duration of just 1 h, after which the gel formed was settled at the bottom. Hence, it can be concluded from the results of the floating study that formulations with a higher HPMC concentration are not be an ideal composition for an in situ gelling system. Among all the formulations, F1, F2, and F3 exhibited desirable floating on the surface of the medium [17]. Photographs of the in vitro floating behavior of the oral raft-forming in situ gelling system of the NTB formulations is shown in Figure 3.

### 3.9. In Vitro Drug Release

A combination of different polymer (HPMC K4M and sodium alginate) concentrations was used to sustain the drug release from the prepared in situ gel formulations. In vitro drug release profile studies were performed on all formulations (F1–F9), as shown in Figure 4. All prepared formulations showed a sustained drug release. F1 (% HPMC and sodium alginate 1:1) showed nearly 100% drug release, and F2, F3, and F4, showed nearly 85% drug release at 12 h. The in vitro drug release study revealed that as the polymer concentration increased there was a considerable decrease in the rate and extent of drug released from the formulation, which was due to an increase in the density of the polymer matrix as well as an increase in the diffusional path length of the drug molecules. F9 showed the least drug release, at 50.73% at 12 h; this was due to the high concentration of both polymers, which formed thick sol–gel formations that retarded the drug release from the formulation. F5, F6, F7, and F8 showed 74.31%, 55.44%, 69.91%, and 59.16% drug release at 12 h, respectively. In all in situ gel formulations, the slow diminution of gel matrix and thickness throughout the in vitro drug release study was due to gel erosion. Polymer gel erosion caused gradual decreases in gel matrix thickness and in all in situ gel formulations throughout the drug investigations. It was seen throughout the experiment that the gel matrix in the dissolution medium quickly swelled at 6–8 h, followed by an erosion of the gel matrix polymer after 10 or 12 h. F1 showed continuous floating for 24 h and sustained the drug release for up to 12 h; hence, it was chosen as the optimized formulation.

### 3.10. Kinetics of Drug Release Studies

According to the regression coefficients, the kinetics of the dissolution data were well suited to the zero-order, Higuchi matrix, Hix Crow, and KP models (KP) (Table 5). Diffusion, swelling, and erosion were the three most essential rate-control mechanisms for the controlled release formulations. The swelling formulations and diffusion mechanisms comprised the relaxing of polymer chains and water absorption, leading the polymer to swell from glassy to rubbery. The mechanism of drug release from the formulation is indicated by the diffusion exponent n. In the KP equation, if the n value is below 0.45 it indicates the release of Fickian diffusion, and if the n value is between 0.45 and 0.89 it indicates non-Fickian (anomalous) transport; n values above 0.89 indicate a case II transport mechanism. This KP model is used to examine drug release from dosage forms when there is more than one type of drug release mechanism or when the release mechanism is unknown. For all factorial design formulations, the value of the diffusion exponent n was between 0.385 to 0.772. The formulations F3, F5, F6, F8, and F9 were found to have non-Fickian release, whereas F1, F2, F4, and F7 were found to have Fickian release. F2, F2, F4, F5, F7, and F8 were best suited to Peppas release, and F6 and F9 were found to be Hix Crow models, respectively. However, F1 showed zero-order release and F3 showed matrix release, as shown in Table 5. The optimized formulation F1 showed Fickian diffusion with zero-order drug release.

## 4. Data Analysis and Optimization

The effect of independent variables such as the concentration of sodium alginate (A) and HPMC K4M (B) on responses such as floating lag time, percentage water uptake, and percentage drug release was analyzed using 3^2^ factorial designs. When different concentrations of factors were loaded at three levels (high, medium, and low), nine different formulations were obtained from the software. The formulations and their responses are depicted in Table 6.

The obtained results show that the independent variables had a significant impact on the dependent variables selected, such as floating lag time (ranging from 2.4 to 89 min), percentage water uptake (ranging from 7.2 ± 0.11 to 44 ± 0.76 %), and percentage drug release (ranging from 47.9 ± 0.63% to 90.5 ± 0.42%). For given levels of each independent variable, the equation in terms of coded factors can be utilized to made predictions about the response. The high levels of the factors are coded as +1 and the low levels of the factors are coded as −1 by default. By comparing the factor coefficients, the coded equation can be used to determine the relative impact of the components. A positive value in the factorial equation indicates a direct relationship with the independent variable. The particular response and a negative value denote inverse correlation between independent variables, and the response is depicted in Table 7**.**

### 4.1. Impact of Independent Variables on Floating Lag Time Response

Formulations F1, F4, and F7, with the same concentration of sodium alginate but different concentrations of HPMC K4M, showed a variation in floating lag time. It was observed that as the concentration of HPMC K4M increased, floating lag time also increased. The quantitative effect of the formulation factors on the dependent variables are represented in Equation (2) and also shown in Table 7. Factor A (sodium alginate) showed a positive effect on floating lag time, which means that as the concentration of sodium alginate increased, floating lag time also increased. However, factor B (HPMC K4M) showed a negative impact on floating lag time, indicating an inverse relationship between factor and response. The higher the polymer concentration, the more time it took to float on the surface compared to formulations with lower polymer concentrations. The predicted and actual value plots and 3D response surface plots showing the effect of independent variables, i.e., factor A (sodium alginate) and factor B (HPMC K4M), on floating lag time are shown in Figure 5. A drastic difference was observed among the formulations when the total floating time was considered. F1 exhibited floating even after 24 h, F4 showed only 1 h floating before the gel formulation settled at the bottom, and F7 showed floating for less than 2 h. Formulations with a higher HPMC concentration resulted in a hard gel that settled at the bottom after floating for an hour, as seen in F7 and F4. Formulations F1, F2, and F3, which had the same concentrations of HPMC K4M but different concentrations of sodium alginate, showed different floating durations and floating lag times. As the concentrations of sodium alginate increased, the total floating time decreased. The maximum floating duration was observed in the formulation with the lowest concentration of sodium alginate. The effect of the factors on floating lag time is represented by the fitted linear regression given in Equation (2) by the software. The ANOVA results for predicting floating lag time are shown in Table 8.
Floating lag time (min) = + 42.43 − 6.94 A[1] + 1.28 A[2] − 38.48 B[1] − 2.10 B[2](2)

### 4.2. Impact of Independent Variables on Percentage Water Uptake Response

From the results obtained for percentage water uptake study, it was observed that all formulations behaved differently at a different time interval. At 60 min, the percentage water uptake for F1, F2, and F3 was 33, 23.5, and 16%, respectively, showing that % water uptake decreased with increasing sodium alginate concentration. At 120 min, the % water uptake by F1 increased to 44% and F3 increased to 42%, but for F2 there was no increase in % water uptake. When formulations F1, F4, and F7, which had had the same concentrations of sodium alginate and different concentrations of HMPMC K4M, were compared with each other, it was observed that as the amount of HPMC K4M increased from 1 to 2 g, the percentage water uptake decreased from 44% to 10% at 120 min. Also evident from the results is that formulations F3, F6, and F9 showed 42, 15, and 7.2% water uptake, respectively, at 120 min. The ANOVA results for predicting % water uptake are shown in Table 8. The percentage water uptake study results of the oral raft-forming in situ gelling system of the NTB formulations are shown in Table 9. Both factors, sodium alginate and HPMC K4M, had negative effects at higher concentrations on percentage water uptake: As the concentration of factors increased, % water uptake decreased, as shown in Table 6 and Table 7. However, both factors showed a positive effect at a lower level of concentration, as seen in Equation (3). Predicted and actual value plots and 3D response surface plots showing the effect of independent variables, i.e., factor A (sodium alginate) and factor B (HPMC K4M), on percentage water uptake at 2 h are shown in Figure 6. Hence, from the noted results and coefficient table, it was evident that HPMC K4M had negative effect on % water uptake. The effect of factors on % water uptake is represented by the fitted linear regression in Equation (3) given by the software.
% Water uptake at 2 h = + 20.84 + 1.76 A[1] − 2.54 A[2] + 10.09 B[1] − 5.51 B[2](3)

**Impact of independent variables on % drug release response at 6 h and at 12 h.** In the developed oral raft-forming in situ gel formulation, the gelling agents employed were sodium alginate and HPMC K4M. The gel formation takes place at an acidic pH; when the formulation is administered orally the sol-to-gel transition occurs, and due to the release of CO_2_ because of the presence of sodium bicarbonate in the formulation, it helps the gel formed by the polymer to float on the surface of the gastric medium, thereby preventing direct contact of the drug with the mucosal layer and leading to sustained release of the drug. The % drug release responses at 6 h and at 12 h were used to identify the effect of factors A and B at three different levels. From the results obtained by the software and Equations (4) and (5), it was observed that both the factors had a positive impact on the % drug release at 6 h and at 12 h. However, factor A at its lowest concentration showed a negative effect on the response. The predicted and actual value plots and 3D response surface plots showing the effect of independent variables, i.e., factor A (sodium alginate) and factor B (HPMC K4M), on percentage drug release at 6 h and 12 h are shown in Figure 7.

The ANOVA results for predicting % drug release at 6 h and 12 h are shown in Table 8. The fitted linear regression equation showing a significant effect on the % drug release response at 6 h and 12 h are shown below.
% Drug release at 6 h = + 49.823.09 A[1] + 12.01 A[2] + 3.78 B[1] + 6.68 B[2](4)
% Drug release at 12 h = + 73.96 − 3.10 A[1] + 10.75 A[2] + 4.35 B[1] + 3.05 B[2](5)

**Optimization:** The effect of various levels of independent variables on the responses can be analyzed by desirability and optimization approaches. Constraints were applied to the dependent variables to achieve an optimized formula by generating desirability plots, as shown in Figure 8. 

Floating lag time (3 min), percentage water uptake (30%), and percentage drug release at 6 h (45%) and at 12 h (85%) were fixed as constraints. The recommended concentration of independent variables A (sodium alginate) and B (HPMC K4M) with a desirability factor near 1.0 (0.822) provided by the software were chosen, which were 3 g of factor A and 1 g of factor B. The final composition of the oral raft-forming in situ gelling system of NTB contained a 100 mg equivalent amount of NTB PEG solid dispersion, 3 g of sodium alginate, 1 g of HPMC K4M, 1% gas-forming agent (sodium bicarbonate) to retain the fluidity of the formulation before administration, 1% sodium citrate, a preservative methyl paraben, and a distilled water vehicle. The given optimized formulation was prepared and evaluated for the responses of floating lag time, percentage water uptake, and percentage drug release at 6 h and at 12 h. The results were compared with the model prediction shown in Table 10. The observed experimental values were amazingly close to the model predicted value. These results are attributed to the validity and reliability of the optimization technique used in the present study using factorial design.

## 5. Conclusions

The oral raft-forming in situ gelling system of anticancer drug NTB was successfully developed using 3^2^ factorial designs. Among various solid dispersions prepared using PEG 6000, the formulation with an NTB:PEG ratio of 1:3 showed the highest solubility of 132.11 ± 2.14 ug/mL. FT-IR spectra and a SEM micrograph revealed an NTB PEG solid dispersion formation. FT-IR spectra revealed no specific interaction between the polymer and the drug. The percentage drug content in the NTB solid dispersion ranged from 96.22 ± 1.67% to 97.70 ± 1.89%. The developed in situ gel formulations exhibited a pH value near 7. The in vitro gelation of an in situ gel formulation showed immediate gelation, and the gel was retained for an extended period of time. From the obtained results of the three-level (3^2^) factorial design analyzing the impact of two independent variables viz. sodium alginate [A] and HPMC K4M [B], it was evident that both the selected factors had a significant effect on the chosen responses, such as floating lag time, water uptake (%), and percentage release of drug, supporting the precision of the design employed for optimization. Thus, the developed oral raft-forming in situ gelling system of NTB may be a favorable and alternative strategy to enhance gastric retention and sustained release of the drug by letting it remain floating in the stomach, thereby augmenting the therapeutic efficacy of NTB.

## Figures and Tables

**Figure 1 polymers-14-02520-f001:**
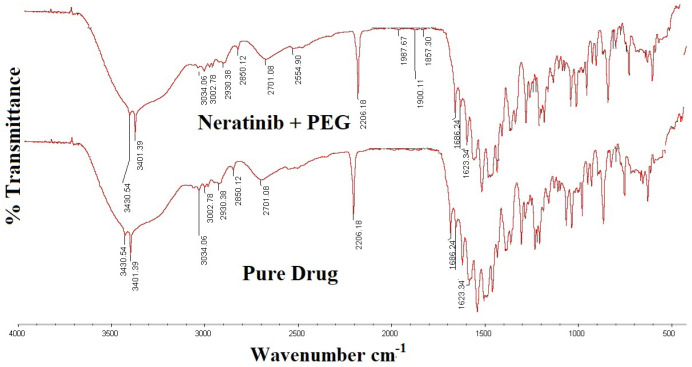
Overlain FT-IR spectra of pure drug neratinib and neratinib–PEG solid dispersion.

**Figure 2 polymers-14-02520-f002:**
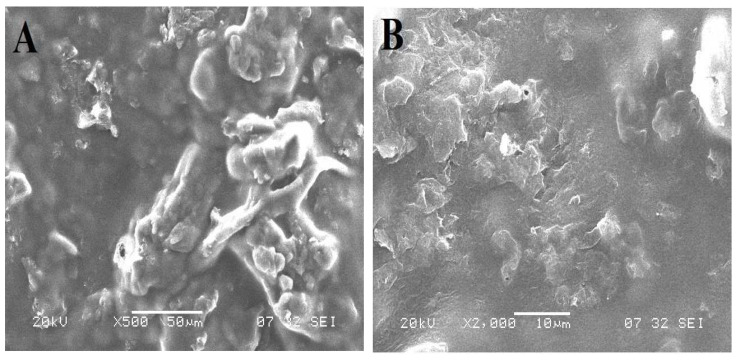
Scanning electron micrographs of (**A**) NTB (pure drug) at 500× magnification and (**B**) NTB–PEG solid dispersion at 2000× magnification.

**Figure 3 polymers-14-02520-f003:**
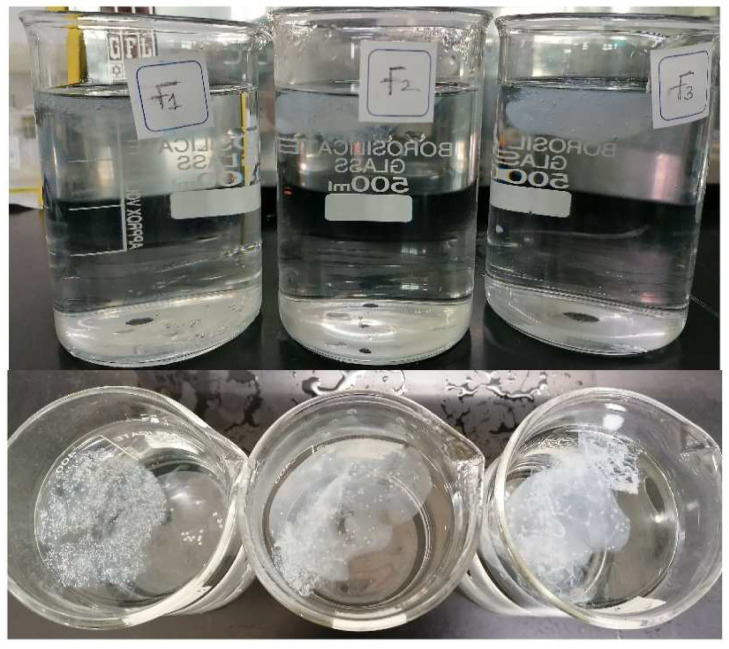
Photographs of formulations F1–F3 depicting the in vitro floating behavior of the oral raft-forming in situ gelling systems of NTB.

**Figure 4 polymers-14-02520-f004:**
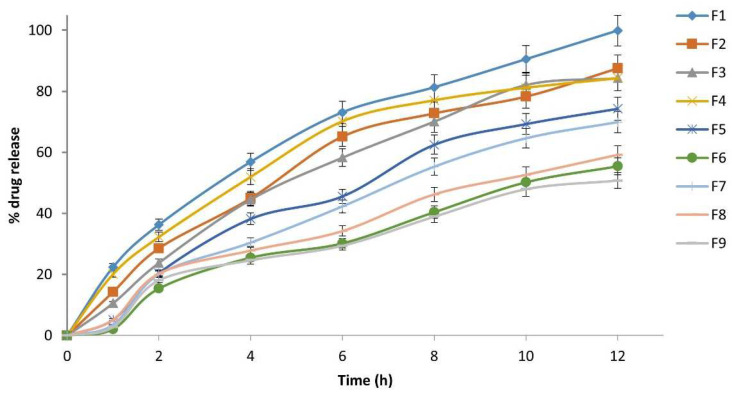
In vitro drug release profile of the oral raft-forming in situ gelling systems of NTB viz. formulations (F1–F9).

**Figure 5 polymers-14-02520-f005:**
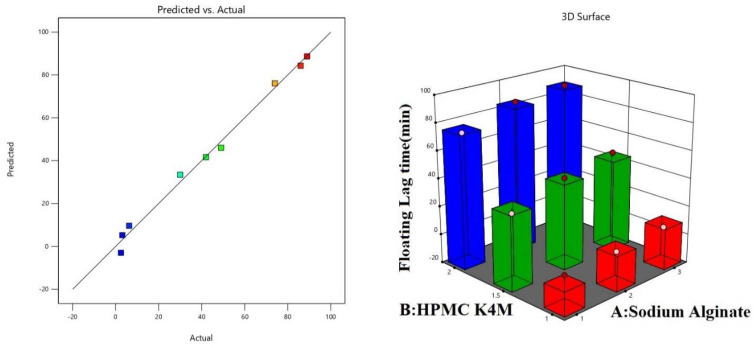
Predicted and actual value plots and 3D response surface plots showing the effect of independent variables, i.e., factor A (sodium alginate) and factor B (HPMC K4M), on response floating lag time.

**Figure 6 polymers-14-02520-f006:**
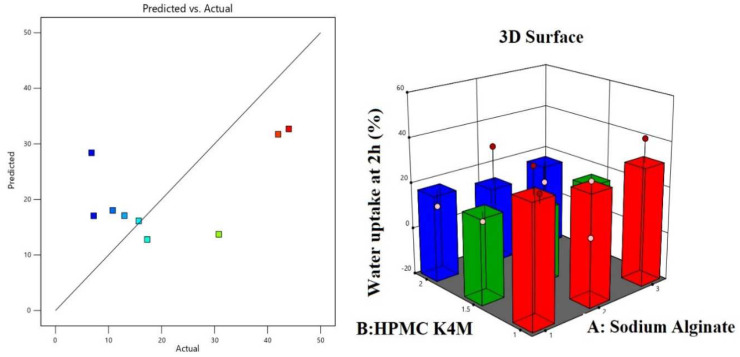
Predicted and actual value plots and 3D response surface plots showing the effect of independent variables, i.e., factor A (sodium alginate) and factor B (HPMC K4M), on % water uptake response at 2 h.

**Figure 7 polymers-14-02520-f007:**
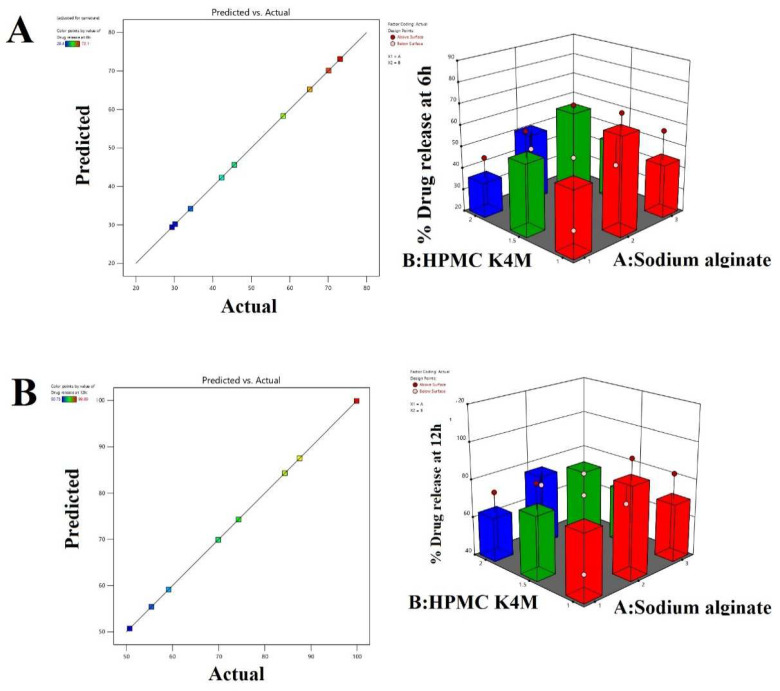
Predicted and actual value plots and 3D response surface plots showing the effect of the independent variables, i.e., factor A (sodium alginate) and factor B (HPMC K4M), on response % drug release at 6 h (**A**) and % drug release at 12 h (**B**).

**Figure 8 polymers-14-02520-f008:**
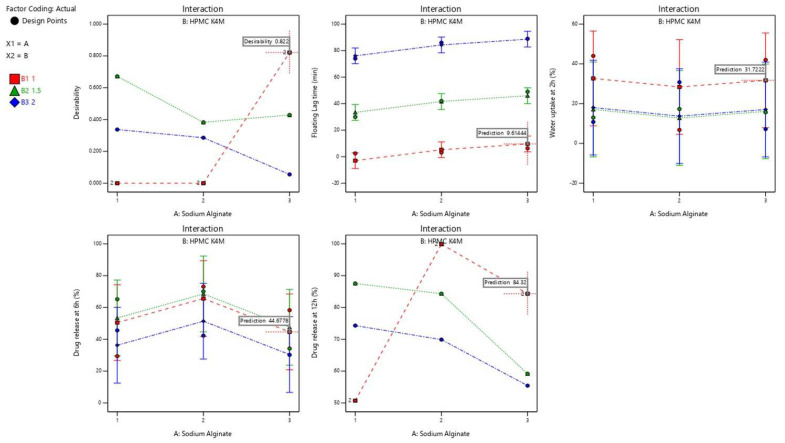
Optimization of the oral raft-forming in situ gelling system of NTB, represented by desirability plots and interactions.

**Table 1 polymers-14-02520-t001:** 3^2^ factorial design for the oral raft-forming in situ gelling system of neratinib, an anticancer drug, obtained using Design-Expert software (version 130.2.0) from Stat-Ease Inc., Minneapolis, MN, USA.

Ingredients	F1	F2	F3	F4	F5	F6	F7	F8	F9
Neratinib (mg)	100	100	100	100	100	100	100	100	100
Sodium alginate (g)	1	2	3	1	2	3	1	2	3
HPMC K4M (g)	1	1	1	1.5	1.5	1.5	2	2	2
Sodium bicarbonate (%)	1	1	1	1	1	1	1	1	1
Sodium citrate (%)	1	1	1	1	1	1	1	1	1
Methyl paraben (%)	0.8	0.8	0.8	0.8	0.8	0.8	0.8	0.8	0.8

**Table 2 polymers-14-02520-t002:** Composition of independent variables and their levels for the preparation of the oral raft-forming in situ gelling system of NTB, an anticancer drug.

Variables	Actual Value (g)			Coded Value		
	Low	Medium	High	Low	Medium	High
A: sodium alginate	1	2	3	−1	0	+1
B: HPMC K4M	1	1.5	2	−1	0	+1

**Table 3 polymers-14-02520-t003:** Percentage drug content and saturation solubility of NTB PEG solid dispersion.

Formulation	Drug: PEG Ratio	Percentage Drug Content NTB Solid Dispersion (%) *	Saturation Solubility of Solid Dispersion (μg/mL) *
Neratinib	-	-	58.3 ± 1.82
A1	1:1	96.22 ± 1.67	96.37 ± 1.67
A2	1:2	97.70 ± 1.89	92.81 ± 1.53
A3	1:3	96.90 ± 2.14	132.11 ± 2.14

* Mean ± SD, *n* = 3.

**Table 4 polymers-14-02520-t004:** Evaluation parameters of the oral raft-forming in situ gelling system of NTB.

Formulations	pH	Drug Content * (%)	In Vitro Gelation	Floating Lag Time * (min)	Total Floating Time (h)	% Drug Release * (6 h)	% Drug Release * (12 h)
F1	7.2 ± 0.44	98.7 ± 0.52	+++	2.47 ± 0.4	>24	73.1 ± 0.42	99.89 ± 0.27
F2	7.8 ± 0.24	99.3 ± 0.32	+++	3.12 ± 0.2	>12	65.2 ± 0.53	87.54 ± 0.65
F3	7.8 ± 0.32	97.8 ± 0.41	+++	6.27 ± 0.2	>12	58.3 ± 0.21	84.32 ± 0.14
F4	7.1 ± 0.21	97.4 ± 0.22	+++	30 ± 0.8	1.0	70.2 ± 0.02	84.33 ± 0.17
F5	7.4 ± 0.52	98.8 ± 0.16	+++	42 ± 0.5	>12	45.6 ± 0.62	74.31 ± 0.02
F6	7.3 ± 0.12	99.2 ± 0.51	+++	49 ± 0.4	>12	30.2 ± 0.76	55.44 ± 0.87
F7	7.1 ± 0.52	97.8 ± 0.46	+++	74 ± 0.6	<3	42.3 ± 0.29	69.91 ± 0.54
F8	7.2 ± 0.52	98.4 ± 0.31	+++	86 ± 0.2	<3	34.2 ± 0.71	59.16 ± 0.22
F9	7.2 ± 0.52	97.7 ± 0.61	+++	89 ± 0.7	<3	29.4 ± 0.63	50.73 ± 0.87

* Mean ± SD, *n* = 3.

**Table 5 polymers-14-02520-t005:** Kinetic studies of the dissolution profile of NTB matrix tablets (values of R^2^, k, and *n*) and mechanism of drug release.

Formulation	Zero Order		Hix Crow		Higuchi Matrix		1st Order		Korsmeyer–Peppas			Mechanism of Drug Release	Release Kinetic
	R^2^	K	R^2^	K	R^2^	K	R^2^	K	R^2^	K	*n*		
F1	0.994	10.845	0.968	25.480	0.979	13.179	0.971	11.194	0.982	10.052	0.385	fickian	Zero order
F2	0.918	10.539	0.973	15.678	0.871	12.802	0.945	11.986	0.996	12.983	0.417	fickian	Peppas release
F3	0.974	6.132	0.835	10.605	0.997	19.769	0.994	16.897	0.987	18.529	0.564	Nonfickian	Higuchi matrix
F4	0.942	12.195	0.899	11.481	0.934	11.983	0.992	10.631	0.995	14.777	0.399	fickian	Peppas release
F5	0.891	10.523	0.971	10.771	0.909	16.911	0.919	12.752	0.979	11.650	0.672	Nonfickian	Peppas release
F6	0.917	15.225	0.993	17.232	0.781	17.668	0.984	11.981	0.843	21.811	0.685	Nonfickian	Hix Crow
F7	0.932	13.809	0.789	10.765	0.927	21.286	0.912	10.098	0.998	16.659	0.391	fickian	Peppas release
F8	0.874	17.220	0.985	13.723	0.801	10.096	0.951	13.231	0.992	10.526	0.772	Nonfickian	Peppas release
F9	0.909	10.448	0.987	11.791	0.923	11.776	0.926	11.875	0.975	15.433	0.566	Nonfickian	Hix Crow

**Table 6 polymers-14-02520-t006:** Observed responses in 3^2^ full factorial design for the oral raft-forming in situ gelling system of NTB.

Formulations	Variables		Responses			
	A (Sodium Alginate) g	B (HPMC K4M) g	Floating Lag Time * (min)	Water Uptake at 2 h *(%)	% Drug Release * (6 h)	% Drug Release * (12 h)
F1	1	1	2.47 ± 0.4	44 ± 0.76	73.1 ± 0.42	99.89 ± 0.27
F2	2	1	3.12 ± 0.2	6.8 ± 0.42	65.2 ± 0.53	87.54 ± 0.65
F3	3	1	6.27 ± 0.2	42 ± 0.71	58.3 ± 0.21	84.32 ± 0.14
F4	1	1.5	30 ± 0.8	13 ± 0.18	70.2 ± 0.02	84.33 ± 0.17
F5	2	1.5	42 ± 0.5	17.3 ± 0.22	45.6 ± 0.62	74.31 ± 0.02
F6	3	1.5	49 ± 0.4	15.7 ± 0.32	30.2 ± 0.76	55.44 ± 0.87
F7	1	2	74 ± 0.6	10.82 ± 0.43	42.3 ± 0.29	69.91 ± 0.54
F8	2	2	86 ± 0.2	30.87 ± 0.21	34.2 ± 0.71	59.16 ± 0.22
F9	3	2	89 ± 0.7	7.2 ± 0.11	29.4 ± 0.63	50.73 ± 0.87

* Mean ± SD, *n* = 3.

**Table 7 polymers-14-02520-t007:** Multiple regression output for dependent variables, showing the intercept, relationship between the factor and variables, and *p*-value obtained from the software.

	Intercept	A[1]	A[2]	B[1]	B[2]	R^2^
**Floating lag time**	+42.4289	−6.93889	1.27778	−38.4756	−2.09556	0.9924
***p*−value**		0.0530	0.0530	<0.0001	<00001	
**% Water uptake at 2 h**	+20.8444	1.75556	−2.5444	10.0889	−5.511	0.9111
***p*−value**		0.059	0.059	0.0517	0.0517	
**% Drug release at 6 h**	+49.82	−3.09	+12.01	+3.78	+6.68	0.8977
***p*−value**		0.0748	0.0748	0.0197	0.0197	
**% Drug release at 12 h**	+73.96	−3.10	+10.75	+4.35	+3.05	0.9312
***p*−value**		0.0523	0.0523	0.0528	0.0528	

**Table 8 polymers-14-02520-t008:** ANOVA results for predicting floating lag time (min), % water uptake, and % drug release at 6 h and at 12 h.

Source	b-Coefficient	Sum of Sqaure	d.f.	Mean Square	F-Value	*p*-Value
Floating Lag Time (min)						
Model	+42.43	9637.8	4	2409.45	131.23	0.0002
A[1]	−6.94	245.49	2	122.74	6.69	0.0530
B[1]	−38.48	9392.32	2	4696.16	255.78	<0.0001
Residual		73.44	4	18.36		
		9711.25	8			
% Water Uptake						
Model	+20.84	489.88	4	122.47	41.59	0.059
A[1]	+1.76	30.54	2	15.27	5.18	0.095
B[1]	+10.09	459.34	2	229.67	7.8	0.051
Residual		1177.84	4	294.46		
		1667.72	8			
% Drug Release at 6 h						
Model	+49.82	2058.12	4	514.53	8.78	0.0292
A[1]	−3.09	622.67	2	311.33	5.31	0.748
B[1]	+3.78	1435.45	2	717.72	12.25	0.0197
Residual		234.41	4	58.6		
		2292.53	8			
% Drug Release at 12 h						
Model	+73.96	800.58	4	200.53	8.78	0.0571
A[1]	−3.10	551.24	2	275.33	5.31	0.852
B[1]	+4.35	249.45	2	124.72	12.25	0.0432
Residual		1389.41	4	347		
		2189.53	8			

**Table 9 polymers-14-02520-t009:** Percentage water uptake study results of the oral raft-forming in situ gelling system of NTB.

Formulations	% Water Uptake *		
	At 30 min	At 60 min	At 120 min
F1	7.5 ± 0.61	33 ± 0.49	44 ± 0.76
F2	15.6 ± 0.31	23.5 ± 0.28	6.8 ± 0.42
F3	16 ± 0.22	16 ± 0.12	42 ± 0.71
F4	7.4 ± 0.34	7.5 ± 0.18	13 ± 0.18
F5	0 ± 0.76	15.3 ± 0.08	17.3 ± 0.22
F6	4.5 ± 0.28	14.79 ± 0.32	15.7 ± 0.32
F7	0.57 ± 0.11	2.2 ± 0.63	10.82 ± 0.43
F8	5.26 ± 0.37	12.5 ± 0.42	30.87 ± 0.21
F9	13 ± 0.28	5.6 ± 0.28	7.2 ± 0.11

* Mean ± SD, *n* = 3.

**Table 10 polymers-14-02520-t010:** Comparison of the observed and predicted values of the independent variables given by the software for the optimized formulation.

Factor A	Factor B	Optimized Formulation Independent Variables	Predicted Value	Observed Value	Desirability
3 g	1 g	Floating lag time (min)	9.61	8.91	0.822
		% Water uptake at 2 h	31.72	30.96	
		% Drug release at 6 h	44.67	43.21	
		% Drug release at 12 h	84.32	86.12	

## Data Availability

The data used to support the findings of this study are available from the corresponding author upon request.

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
