# Peer review of "Development and Characterization of Oral Raft Forming In Situ Gelling System of Neratinib Anticancer Drug Using 32 Factorial Design"

_polymers, 2022, doi:10.3390/polym14132520_

Round 1

Reviewer 1 Report

pH of 1.2 seems to be very high if the purpose is to mimic the stomach pH

What is the saturation solubility of free neratinib in aqueous solution?

Line 192, missing citation

Figure 2. A and B are taken with different magnifications.

Figure legend of each figure is rather simple. For figure 3, it is confusing to understand what is presented in the figure. The physical appearance from F1-F3 is very different. It almost looks as if something is not dissolved in the gel, especially F1. Please clarify.

Section 3.9. Half of the paragraph is italic, while the other is not. Section 3.10 had the similar issue.

Author Response

The authors are highly thankful to esteemed reviewers for their excellent comments for further improvement of the article. Responses to queries rose by esteemed reviewers are given below in point by point manner. All the revisions have been made bold in the revised manuscript.

Reviewer 2 Report

An intersting work about the development and characterization of oral raft forming in situ gelling system of neratinib anticancer drugs by usinf a factorial design.

However, some  improvements are needed:

1. The dimensions of the two sieves ("sieve no 22" and "sieve #60"), used in the article, should be specified.

2. At section 2.3.6. Water uptake study, you must introduce the reference which describes the method reported by Jafar et al. From my point of view, the best method to remove the excess medium embedded in the gel is the centrifugation.

3. You used SEM for demonstrating the cristalline state of pure neratinib and amorphous state of PEG-Neratinib solid dispersion. From my point of view, a much more credible method, for studying the cristallinity, would be X-ray diffraction. At least, improve the resolution of the two pictures of Fig. 2!

4. The same request for Fig.1!

Author Response

The authors are highly thankful to esteemed reviewers for their excellent comments for further improvement of the article. Responses to queries rose by esteemed reviewers are given below in point by point manner. All the revisions have been made and highlighted in the revised manuscript.
